# Prevalence of human schistosomiasis in various regions of Tanzania Mainland and Zanzibar: A systematic review and meta-analysis of studies conducted for the past ten years (2013–2023)

**Nicolaus Omari Mbugi** [1,2]*, **Hudson Laizer**[2]*, **Musa Chacha**[3]*, **Ernest Mbega**[1]*

**1** The Nelson Mandela African Institution of Science and Technology, School of Life Sciences and Bioengineering, Arusha, Tanzania, **2** Mbeya University of Science and Technology, College of Science and Technical Education, Mbeya, Tanzania, **3** Arusha Technical College, Arusha, Tanzania

* nicolausm@nm-aist.ac.tz (NOM); hudson.laizer@must.ac.tz (HL); musa.chacha@nm-aist.ac.tz (MC); ernest.mbega@nm-aist.ac.tz (EM)

**Data Availability Statement:** Data are provided as part of the submitted article (Table 1)

## Abstract

Schistosomiasis is a significant public health problem in Tanzania, particularly for the people living in the marginalized settings. We have conducted a systematic review with meta-analysis on the prevalence of schistosomiasis to add knowledge towards the development of effective approaches to control the disease in Tanzania. Online databases namely, Pub Med, SCOPUS and AJOL, were systematically searched and a random effect model was used to calculate the pooled prevalence of the disease. Heterogeneity and the between studies variances were determined using Cochran (Q) and Higgins ($I^2$) tests, respectively. A total of 55 articles met the inclusion criterion for this review and all have satisfactory quality scores. The pooled prevalence of the disease in Tanzania was 26.40%. Tanzania mainland had the highest schistosomiasis prevalence (28.89%) than Zanzibar (8.95%). Sub-group analyses based on the year of publication revealed the going up of the pooled prevalence, whereby for (2013–2018) and (2018–2023) the prevalence was 23.41% and 30.06%, respectively. The prevalence of the *Schistosoma mansoni* and *Schistosoma hematobium* were 37.91% and 8.86% respectively. Mara, Simuyu, and Mwanza were the most prevalent regions, with a pooled prevalence of 77.39%, 72.26%, and 51.19%, respectively. The pooled prevalence based on the diagnostic method was 64.11% for PCR and 56.46% for POC-CCA, which is relatively high compared to other tests. Cochrans and Higgins ($I^2$) test has shown significant heterogeneity (p-value = 0.001 and $I^2$ = 99.6). Factors including age, region, diagnostic method and sample size have shown significant contribution to the displayed heterogeneity. The pronounced and increasing prevalence of the disease suggests potential low coverage and possibly lack of involvement of some regions in the control of the disease. This, therefore, calls for an intensive implementation of control interventions in all endemic regions, preferably using an integrated approach that targets several stages of the disease lifecycle.

**Funding:** The author(s) received no specific funding for this work.

**Competing interests:** The authors have declared that no competing interests exist.

## Author summary

Schistosomiasis is a devastating tropical and sub-tropical disease that disproportionately infects those in resource-limited settings, which causes death, morbidity and socioeconomic impact. The disease is caused by blood parasites under the genus *Schistosoma*. Tanzania is one of the schistosomiasis burdened countries in the sub-Saharan region. Praziquantel mass drug administration has been a predominant schistosomiasis control strategy in the country. We have conducted a systematic review and meta-analysis using a prevalence dataset from published literature, aiming at assessing the country's schistosomiasis burden (prevalence) following several years of disease control. We further conducted a subgroup analysis to assess the contributing factors to the observed disease prevalence. The obtained information provides insight into the impact of schistosomiasis control in Tanzania, and highlights the potential for positive change and improvement in the country's fight against this disease.

## 1. Introduction

Schistosomiasis is one of the most pervasive tropical and sub tropical neglected diseases caused by blood helminths that belong to the genus *Schistosoma* [1,2]. This genus comprises several species that infect a vast range of hosts; however, only six are of clinical importance, namely *Schistosoma mansoni*, *Schistosoma japonicum*, *Schistosoma mekongi*, *Schistosoma guineansis*, *Schistosoma intercalatum* and *Schistosoma haematobium* [3]. The first five species cause intestinal schistosomiasis, whereas the latter causes urogenital schistosomiasis. Schistosomiasis is of public health importance due to its socioeconomic impact, ranking second after Malaria [4]. Recent data shows that over 237 million people are infected by schistosomiasis on a global scale, consequently accounting for 280,000 mortalities yearly [5,6,7]. Of all continents, Africa particularly the Sub-Saharan region, is leading in terms of disease burden, carrying 90% of all global cases [6]. In the region, Tanzania marks the second country in terms of disease burden, only surpassed by Nigeria [8,9]. Typically, *Schistosoma mansoni* and *Schistosoma hematobium* causing intestinal and urogenital schistosomiasis respectively, are endemic species in Tanzania, which cumulatively contribute to the pronounced prevalence [10]. Tanzania is the union of two countries, Zanzibar and Tanganyika, which was later referred to as Tanzania mainland. These two parts of the country have different backgrounds regarding schistosomiasis control. In Zanzibar, the history of schistosomiasis control dates back to the 1980s, when the disease prevalence, particularly urogenital schistosomiasis, was high [11]. The control of the disease was enforced through praziquantel mass drug administration in schools. This initiative was further enhanced by the Zanzibar Elimination of Schistosomiasis Transmission (ZEST) project between 2011 and 2017 [12, 13]. During the project, biannual praziquantel mass drug administration was implemented in both Pemba and Unguja islands to schools and the community at large. In addition, over the course of the project, other control interventions including biocontrol of snails and behavioural changes were also employed [12]. Since the inception of the mentioned control interventions, there has been a considerable decrease in the disease burden from above 50% to 5% in 2020 in both islands. In view of this, current control initiatives focus on the complete elimination of the disease to make Zanzibar amongst the few regions in sub-Saharan Africa to achieve interruption of disease transmission [11].

Meanwhile, schistosomiasis control in Tanzania's mainland dates back to 2000s, when the Schistosomiasis and Soil-transmitted Helminths Control Programme (NSSCP) was founded

[14]. NSSCP is a joint partnership between the Ministry of Health, Community Development, Gender, Elderly and Children (MoHCDGEC) and the Ministry of Education and Vocational Training (MoEVT) established with the support of the Schistosomiasis Control Initiative (SCI). The programme has done much on disease surveillance and control through mass drug administration (MDA) using praziquantel. The praziquantel MDA mainly focused on school children, and from 2009 to 2018, more than 33.3 million drugs were delivered to the mentioned group [15]. The program covers all administrative regions that are endemic to schistosomiasis (about 17 regions), with a particular focus on the regions in a northern-western zone, which are highly endemic to the disease [15, 16]. In the regions that have prevalence > 50, the program was further extended to other high-risk groups [16]. However, the disease prevalence is still very high across endemic areas in the country, ranging from 12.7% to 87.6% [14]. The disease is more abundant in societies living along the shoreline of Lake Victoria, particularly in the Mwanza region, with vulnerable groups being pre-school and school-aged children as well as women[5]. The vulnerability of the aforementioned groups is attributed to their routine domestic activities that potentially expose them to infected water [7].

Both forms of schistosomiasis (urogenital and intestinal) are acquired through skin penetration of infective larvae (cercariae) into susceptible hosts[4]. This occurs when a susceptible host is exposed to infected water, and hence, infective larvae penetrate and migrate to their resident sites, where they grow into adult worms [3,4]. In the host body, adult worms reside in various destinations specific to the schistosome species, where they copulate and lay eggs [3,4]. The laid eggs are responsible for the pathophysiology of the disease, which is then clinically manifested in two forms; the acute and chronic manifestations [3,4]. Summarily, clinical manifestations of urogenital schistosomiasis include dysuria, nutritional deficiencies, haematuria, hydronephrosis, urinary bladder squamous cell carcinoma, and urinary bladder lesions [5]. For intestinal schistosomiasis, medical conditions such as splenomegaly, hepatomegaly, and progressive periportal fibrosis are manifested [5].

Considering the WHO guideline, the control of schistosomiasis in Tanzania and elsewhere predominantly relied on preventive chemotherapy by using Praziquantel drug. However, there is a need to assess the impact of this control intervention on the disease burden over the years of its implementation. Therefore, the present systematic review compiles epidemiological data on the prevalence of schistosomiasis in both Tanzania mainland and Zanzibar from 2013 to 2023. This will provide baseline information regarding the country's response to Praziquantel mass drug administration as the mainstay for disease control, as well as provide the highlights for potential improvements in the fight against schistosomiasis.

## 2. Methodology

### 2.1 Data acquisition

Computer-assisted searches in online databases including Pub Med, SCOPUS and AJOL were done by using advanced search options to obtain relevant articles. MeSH term options from PubMed were usedf to obtain some terminologies used in the material search. Whereby, the search term used were Schistosomiasis OR Bilharzia* OR Schistosoma* OR Katayama* AND Epidemiology OR Prevalence AND Tanzania OR Zanzibar in combination. The obtained articles were primarily screened based on the relevance of their titles and abstracts to the reviewed topic by applying the prior defined inclusion and exclusion criterion. Duplicates were also removed using Mendeley. Again, full screenings were further performed to obtain the final set of publications used to compose the present review article.

## 2.2 Exclusion and inclusion criteria

Publications excluded from the use in the present review include publications not written in the English language, titles not relevant to the reviewed topic, reporting prevalence dataset from countries other than Tanzania, duplicating results of the research work from large project or research group, non-human studies, containing pooled datasets from two countries or more, contain datasets from a prospective longitudinal study following mass drug administration, as well as articles assessing genetic dynamics among both *S. hematobium* and *S. mansoni* populations. Whereas published articles containing prevalence data from the Tanzanian population and not returning visitors from Tanzania collected within the past 10 years (2013 to 2023), reporting baseline data from the intervention studies, prevalence data of schistosomiasis in various regions of Tanzania mainland and Zanzibar, assessing sensitivity of various diagnostic techniques with a clearly defined sample size and number of cases were included.

## 2.3 Articles selection and data extraction

Following combining results from the searched online databases, author NM and HL scanned all retrieved publications based on the titles and abstracts for their eligibility to be fully reviewed. Eligible articles were full text reviewed by two independent authors (NM and HL) for their inclusion illegibility in data extraction. Articles from both authors that are matching were selected for data extraction. For the case of any mismatch other two authors (MC and EM) were consulted. Prior to data extraction, all duplicates were removed. Afterwards, data were independently abstracted by two authors (NM and HL) using a pre-designed form in Microsoft Excel. The form was pre-tested by all authors (NM, HL, MC, EM) prior to its use in data extraction. On the occasion that there was a difference in data extracted by the two authors from similar articles, other two authors were invited (MC and EM) to extract the data independently. Similar data extracted by at least three of the four authors was taken. Extracted data includes the author's name, year of publication, study regions, diagnostic method used, sample size (study population), number of cases, target groups, study design used as well as the age range (Table 1).

## 2.4 Assessment of study quality and risk of bias

Quality and risk of bias for the selected publications were evaluated on the basis of criteria stipulated in the Joanna Briggs Institute critical appraisal checklist for use in reviews of prevalence studies[17]. Briefly, each criterion was accredited 1 grade if a criterion was met and 0 if a criterion was not met. Nine (9) was the maximum score given when all criteria were met, whereas 0 was the minimum score given to an article when none of the criteria were met. Articles scored cumulative grades ranging from 0–4 were regarded are of low quality, 5–7 moderate quality and 8–9 high quality. After that, articles demonstrating moderate to high quality were included in the present systematic review and meta-analysis. Authors (NM and HL) independently evaluated the quality of included publications.

## 2.5 Meta-analysis

Meta-analysis was done in line with the previously published protocol by employing a random effects model [17]. Through this, point estimates of the weighted prevalence of the datasets from the included studies were computed at 95% confidence intervals (CIs) and presented by forest plot. Heterogeneity, as well as variation between studies was assessed using Cochran's Q (chi-square) and (Higgins) $I^2$ tests, respectively. $I^2$ values above 50% were considered to indicate substantial heterogeneity. For Cochran's Q (chi-square) test, the heterogeneity was

**Table 1. General characteristics of the reviewed articles.**

| Author | Year of publication | Study area | Study design | Targeted species | Targeted group | Age (Years) | Diagnostic methods | Sample size | Cases |
|---|---|---|---|---|---|---|---|---|---|
| Angelo et al.[18] | 2018 | Shinyanga | Longitudinal | S. hematobium | School-aged children | 12 to 14 | Urine filtration | 282 | 98 |
| Bakuza et al.[19] | 2018 | Lindi | Cross-sectional | S. hematobium | Children | 9 to 12 | Urine filtration | 190 | 44 |
| Barda et al.[20] | 2013 | Mwanza | Unspecified | S. mansoni | Children | 4 to 19 | Direct smear | 201 | 8 |
| Barda et al. [20] | 2013 | Mwanza | Unspecified | S. mansoni | Children | 4 to 19 | Kato Katz | 201 | 66 |
| Barda et al. [20] | 2013 | Mwanza | Unspecified | S. mansoni | Children | 4 to 19 | Mini FLOTAC | 201 | 98 |
| Barda et al.[21] | 2014 | Mwanza | Cross-sectional | S. mansoni | School children and adults | Above 5 | Mini FLOTAC | 251 | 58 |
| Barda et al. [21] | 2014 | Mwanza | Cross-sectional | S. mansoni | School children and adults | Above 5 | Formol ether concentration | 251 | 32 |
| Barda et al. [21] | 2014 | Mwanza | Cross-sectional | S. mansoni | School children and adults | Above 5 | Direct smear | 251 | 9 |
| Barda et al. [21] | 2014 | Mwanza | Cross-sectional | S. mansoni | School children and adults | Above 5 | Urine concentration | 151 | 72 |
| Bukindu et al. [22] | 2016 | Mwanza | Cross-sectional | S. mansoni | School-aged children | 8 to 18 | Kato Katz | 625 | 229 |
| Casacuberta et al. [23] | 2016 | Mwanza | Cross-sectional | S. mansoni | School-aged children | 9 to 12 | Kato Katz | 404 | 303 |
| Casacuberta et al. [23] | 2016 | Mwanza | Cross-sectional | S. mansoni | School-aged children | 9 to 12 | POC-CCA | 404 | 172 |
| Fanz et al. [24] | 2023 | Mwanza | Cross-sectional | S. mansoni | Orphans | 6 to 18 | POC-CCA | 144 | 91 |
| Fanz et al. [24] | 2023 | Mwanza | Cross-sectional | S. mansoni | Street children | 6 to 18 | POC-CCA | 122 | 103 |
| Fanz et al.[24] | 2023 | Mwanza | Cross-sectional | S. mansoni | Orphans | 6 to 18 | Kato Katz | 144 | 28 |
| Fanz et al. [24] | 2023 | Mwanza | Cross-sectional | S. mansoni | Street children | 6 to 18 | Kato Katz | 112 | 86 |
| Fulgence et al. [25] | 2023 | Dar es Salaam | Cross-sectional | S. mansoni | University students | 19 to 33 | Kato Katz | 272 | 22 |
| Fuss et al.[26] | 2018 | Mwanza | Cross-sectional | S. mansoni | School-aged children | 7 to 16 | Kato Katz | 297 | 253 |
| Fuss et al. [26] | 2018 | Mwanza | Cross-sectional | S. mansoni | School-aged children | 7 to 16 | POC-CCA | 297 | 282 |
| Fuss et al. [26] | 2018 | Mwanza | Cross-sectional | S. mansoni | School-aged children | 7 to 16 | Real-time PCR | 297 | 276 |
| Fuss et al.[27] | 2020 | Mwanza | Cross-sectional | S. mansoni | Adults | 18 to 70 | Kato Katz | 36 | 12 |
| Fuss et al. [27] | 2020 | Mwanza | Cross-sectional | S. mansoni | Adults | 18 to 70 | POC-CCA | 36 | 23 |
| Fuss et al. [27] | 2020 | Mwanza | Cross-sectional | S. mansoni | Adults | 18 to 70 | Serum real-time PCR | 36 | 27 |
| Fuss et al. [27] | 2020 | Mwanza | Longitudinal | S. mansoni | Adults | 18 to 70 | Urine real-time PCR | 36 | 11 |
| Fuss et al. [28] | 2021 | Mwanza | Cross-sectional | S. mansoni | Adults | 17 to 70 | POC-CCA | 100 | 80 |
| Fuss et al. [28] | 2021 | Mwanza | Cross-sectional | S. mansoni | Adults | 17 to 70 | Serum real-time PCR | 95 | 84 |
| Fuss et al. [28] | 2021 | Mwanza | Cross-sectional | S. mansoni | Adults | 17 to 70 | DBS real-time PCR | 100 | 41 |
| Fuss et al.[28] | 2021 | Mwanza | Cross-sectional | S. mansoni | Adults | 17 to 70 | Kato Katz | 98 | 43 |
| Kaatano et al. [29] | 2015 | Mwanza | Cross-sectional | S. mansoni | Adults | 12 to 85 | Kato Katz | 388 | 121 |
| Kapiga et al. [30] | 2021 | Mwanza and Kagera | Cross-sectional | Schistosoma spp. | Adult | Above 18 | CAA | 1112 | 924 |
| Kayange et al. [31] | 2020 | Mwanza | Cross-sectional | S. mansoni | School-aged children | 6 to 13 | POC-CCA | 507 | 253 |
| Kayange et al. [31] | 2020 | Mwanza | Cross-sectional | S. hematobium | School-aged children | 6 to 13 | Urine filtration | 507 | 8 |
| Keller et al.[32] | 2020 | Zanzibar | Cross-sectional | S. hematobium | Children and adults | 9 to 55 | qPCR(Dra1 DNA) | 792 | 212 |

*(Continued)*

**Table 1.** (Continued)

| Author | Year of publication | Study area | Study design | Targeted species | Targeted group | Age (Years) | Diagnostic methods | Sample size | Cases |
|---|---|---|---|---|---|---|---|---|---|
| Keller et al. [32] | 2020 | Zanzibar | Cross-sectional | *S. hematobium* | Children and adults | 9 to 55 | Urine filtration | 792 | 105 |
| Keller et al. [32] | 2020 | Zanzibar | Cross-sectional | *S. hematobium* | Children and adults | 9 to 55 | Microhaematuria | 792 | 109 |
| Kinug'hi et al. [33] | 2014 | Mwanza | Cross-sectional | *S. mansoni* | Pre-school and school-aged children | 3 to 13 | Kato Katz | 1,546 | 613 |
| Kinug'hi et al. [33] | 2014 | Mwanza | Cross-sectional | *S. hematobium* | Pre-school and school-aged children | 3 to 13 | Nucleopore filtration method | 1,546 | 305 |
| Kinunghi et al. [34] | 2017 | Mara | Cross-sectional | *S. mansoni* | School-aged children | 6 to 15 | Kato Katz | 928 | 794 |
| Kisiringyo et al. [35] | 2020 | Morogoro | Cross sectional | *S. hematobium* | School children | 5 to 16 | Formal-ether sedimentation | 374 | 186 |
| Kisiringyo et al. [35] | 2020 | Morogoro | Cross sectional | *S. mansoni* | School children | 5 to 16 | Formal-ether sedimentation | 374 | 1 |
| Knopp et al. [36] | 2015 | Zanzibar | Unspecified | *S. hematobium* | School-aged children | 8 to12 | Urine filtration | 1740 | 58 |
| Knopp et al. [37] | 2018 | Zanzibar | Cross-sectional | *S. hematobium* | Adults | 20 to 55 | Urine filtration | 18,155 | 490 |
| Knopp et al. [37] | 2018 | Zanzibar | Cross-sectional | *S. hematobium* | Children | 9 to 12 | Urine filtration | 39,207 | 2,117 |
| Masikini et al. [38] | 2019 | Mwanza | Case-control | *S. mansoni* | HIV infected adults | Above 18 | Microscopy | 170 | 19 |
| Masikini et al. [38] | 2019 | Mwanza | Case-control | *S. mansoni* | HIV infected adults | Above 18 | CAA | 188 | 82 |
| Mazigo et al. [39] | 2019 | Mwanza | Cross-sectional | *S. mansoni* | HIV infected children | 1 to 16 | Kato Katz | 103 | 11 |
| Mazigo et al. [39] | 2019 | Mwanza | Cross-sectional | *S. mansoni* | HIV infected children | 1 to 16 | POC-CCA | 134 | 45 |
| Mazigo et al. [40] | 2021 | Mwanza | A prospective longitudinal | *S. mansoni* | School-aged children | 7 to 17 | Kato Katz | 399 | 226 |
| Mazigo et al. [40] | 2021 | Mwanza | A prospective longitudinal | *S. mansoni* | School-aged children | 7 to 17 | POC-CCA | 399 | 398 |
| Mazigo et al. [41] | 2017 | Mara | Cross-sectional | *S. mansoni* | Adults | 18 to 89 | Kato Katz | 412 | 232 |
| Mazigo et al. [42] | 2014 | Mwanza | Cross-sectional | *S. mansoni* | Adults | 21 to 55 | Kato Katz | 1,785 | 854 |
| Mazigo et al.[5] | 2021 | Ruvuma | Cross-sectional | *S. hematobium* | Pre and school-aged children | 1 to 13 | Urine filtration | 1,560 | 13 |
| Mazigo et al. [5] | 2021 | Ruvuma | Cross-sectional | *S. mansoni* | Pre and school-aged children | 1 to 13 | Kato Katz | 1,560 | 236 |
| Mazigo et al. [5] | 2021 | Ruvuma | Cross-sectional | *S. mansoni* | Pre and school-aged children | 1 to 13 | POC-CCA | 574 | 125 |
| Mazigo et al. [43] | 2018 | Mwanza | Prospective longitudinal study | *S. mansoni* | General population | 15 to 55 | Kato Katz | 419 | 242 |
| Mazigo et al. [43] | 2018 | Mwanza | Prospective longitudinal study | *S. mansoni* | General population | 15 to 55 | POC-CCA | 419 | 365 |
| Mazigo et al. [44] | 2018 | Mwanza | Cross-sectional | *S. mansoni* | HIV infected adults | 15 to 55 | Kato Katz | 979 | 463 |

(*Continued*)

**Table 1.** (Continued)

| Author | Year of publication | Study area | Study design | Targeted species | Targeted group | Age (Years) | Diagnostic methods | Sample size | Cases |
|---|---|---|---|---|---|---|---|---|---|
| Mazigo et al. [44] | 2018 | Mwanza | Cross-sectional | *S. mansoni* | HIV infected adults | 15 to 55 | POC-CCA | 979 | 592 |
| Mhimbira et al. [45] | 2017 | Dar es Salaam | Cohort | *S. mansoni* | Children and adults with TB | above 18 | POC-CCA | 597 | 55 |
| Mhimbira et al. [45] | 2017 | Dar es Salaam | Cohort | *S. hematobium* | Children and adults with TB | above 18 | Urine filtration | 597 | 19 |
| Mnkugwe et al. [46] | 2020 | Simiyu | Cross-sectional | *S. hematobium* | School-aged children | 5 to 19 | Kato Katz | 830 | 752 |
| Mohamed et al. [47] | 2018 | Mwanza | Cross-sectional | *S. mansoni* | School-aged children | 9 to 11 | Kato Katz | 327 | 103 |
| Mueller et al. [48] | 2019 | Mwanza | Cross-sectional | *S. mansoni* | General population | 1 to 95 | Kato Katz | 930 | 641 |
| Mueller et al. [48] | 2019 | Mwanza | Cross-sectional | *S. mansoni* | General population | 1 to 95 | POC-CCA | 930 | 879 |
| Mugono et al. [49] | 2014 | Mwanza | Cross-sectional | *S. mansoni* | School-aged children | 4 to 15 | Kato Katz | 773 | 494 |
| Munisi et al.[50] | 2016 | Mara | Cross-sectional | *S. mansoni* | School-aged children | 6 to 16 | Kato Katz | 513 | 431 |
| Mushi et al. [8] | 2022 | Lindi | Cross-sectional | *S. hematobium* | Preschool aged Children | under 5 | Urine filtration | 385 | 65 |
| Mushi et al. [51] | 2022 | Lindi | Cross-sectional | *S. hematobium* | School-aged children | 6 to 17 | Urine filtration | 649 | 342 |
| Ndokeji et al. [52] | 2016 | Mwanza | Cross-sectional | *S. mansoni* | Pre-school and School-aged children | 4 to 14 | Kato Katz | 454 | 363 |
| Ng'weng'weta et al. [53] | 2017 | Dar es Salaam | Cross-sectional | *S. hematobium* | Pre-school children | 6 to 7 | Urine centrifugation | 424 | 8 |
| Ngasala et al. [54] | 2019 | Dodoma | Cross-sectional | *S. hematobium* | School-aged children | 5 to 16 | Microscopy | 353 | 24 |
| Ngasala et al. [54] | 2019 | Zanzibar | Cross-sectional | *S. hematobium* | School-aged children | 7 to 14 | Microscopy | 150 | 58 |
| Ngassa et al. [55] | 2023 | Kilimanjaro | Cross-sectional | *S. hematobium* | Women | 15 to 45 | Sedimentation | 216 | 5 |
| Nkya [56] | 2023 | Morogoro | Cross sectional | *S. hematobium* | School children | 6 to 16 | Microscopy | 884 | 287 |
| Nyundo et al. [57] | 2017 | Dodoma | Cross-sectional | *S. mansoni* | Psychiatric patients | 12 to 69 | Direct wet preparation and formol-ether concentration | 233 | 12 |
| Ogweno et al. [58] | 2023 | Simiyu | Cross-sectional | *S. mansoni* | School-aged children | Unspecified | Kato Katz | 363 | 150 |
| Palmeirim et al. [59] | 2021 | Morogoro | Cross-sectional | *S. mansoni* | School-aged children | 6 to12 | POC-CCA | 427 | 53 |
| Pham et al. [60] | 2023 | Mwanza | Cross-sectional | *Schistosoma spp.* | Adults | Above 18 | CAA | 1,923 | 873 |
| Rite et al. [61] | 2020 | Geita | Cross-sectional | *S. hematobium* | Women | 15 to 49 | Urine filtration | 426 | 19 |
| Rosinger et al. [62] | 2018 | Mwanza and Shinyanga | Cross-sectional | *S. hematobium* | Non-pregnant women | 18 to 50 | Urine filtration | 209 | 12 |
| Rosinger et al. [62] | 2018 | Mwanza and Shinyanga | Cross-sectional | *S. mansoni* | Non-pregnant women | 18 to 50 | Kato Katz | 205 | 11 |
| Ruganuza et al. [63] | 2015 | Mwanza | Cross-sectional | *S. mansoni* | Pre-school children | 1 to 6 | Kato Katz | 400 | 178 |

*(Continued)*

**Table 1.** (Continued)

| Author | Year of publication | Study area | Study design | Targeted species | Targeted group | Age (Years) | Diagnostic methods | Sample size | Cases |
|---|---|---|---|---|---|---|---|---|---|
| Ruganuza et al. [63] | 2015 | Mwanza | Cross-sectional | *S. mansoni* | Pre-school children | 1 to 6 | POC-CCA | 400 | 320 |
| Said et al.[64] | 2017 | Dar es Salaam | Prospective longitudinal | *Schistosoma spp.* | Pre-school aged children | Under 5 | POC-CCA | 308 | 47 |
| Said et al. [64] | 2017 | Dar es Salaam | Prospective longitudinal | *S. hematobium* | Pre-school aged children | Under 5 | Urine filtration | 308 | 3 |
| Samweli et al. [65] | 2023 | Shinyanga | Cross-sectional | *S. mansoni* | Secondary school students | 11 to 20 | Kato Katz | 620 | 12 |
| Shabani et al. [66] | 2022 | Kagera | Cross sectional | *S. mansoni* | Adults | 18 to 55 | Formal-ether sedimentation | 328 | 36 |
| Sikalengo et al. [67] | 2018 | Dar es Salaam | Cohort study | *S. mansoni* | Adult TB patients | Above 18 | Stool microscopy | 460 | 8 |
| Sikalengo et al. [67] | 2018 | Dar es Salaam | Cohort study | *S. mansoni* | Adult TB patients | Above 18 | POC-CCA | 460 | 19 |
| Sikalengo et al. [67] | 2018 | Dar es Salaam | Cohort study | *S. hematobium* | Adult TB patients | Above 18 | Urine filtration | 460 | 16 |
| Sikalengo et al. [67] | 2018 | Morogoro | Cohort study | *S. mansoni* | Adult TB patients | Above 18 | Stool microscopy | 208 | 7 |
| Sikalengo et al. [67] | 2018 | Morogoro | Cohort study | *S. mansoni* | Adult TB patients | Above 18 | POC-CCA | 208 | 34 |
| Sikalengo et al. [67] | 2018 | Morogoro | Cohort study | *S. hematobium* | Adult TB patients | Above 18 | Urine filtration | 208 | 3 |
| Siza et al.[68] | 2015 | Kagera, Mara, Shinyanga and Mwanza | Cross-sectional | *S. mansoni* | School-aged children | 7 to 16 | Kato Katz | 5,952 | 898 |
| Siza et al. [68] | 2015 | Kagera, Mara, Sinyanga and Mwanza | Cross-sectional | *S. hematobium* | School-aged children | 7 to 16 | Urine filtration | 5,826 | 519 |
| Siza et al.[69] | 2015 | Kagera, Mara and Mwanza | Cross-sectional | *S. mansoni* | Adults | Not specified | Kato Katz | 1,606 | 199 |
| Siza et al. [69] | 2015 | Kagera, Mara and Mwanza | Cross-sectional | *S. mansoni* | Adults | Not specified | Urine filtration | 1,400 | 25 |
| Wang et al.[70] | 2019 | Zanzibar | A randomized controlled trial | *S. hematobium* | General population | Unspecified | Urine filtration | 6,000 | 175 |
| Yangaza et al. [71] | 2020 | Dar es salaam | Cross sectional | *S. hematobium* | School children | 7 to 15 | Urine filtration | 250 | 3 |

considered significant at p-value = 0.1. The univariate subgroup analysis to assess the contributing factors to the observed heterogeneity was computed with regard to the year of publication, region, diagnostic method, sample size, age of the participants and schistosoma species. The mentioned factors were grouped in the following order; year of publication into 2 groups; (2013–2018) and (2018–2023), regions into 17 groups; Ruvuma, Lindi, Mwanza, Dar es salaam, Morogoro, Mara, Simiyu, Shinyanga, Geita, Zanzibar, Kilimanjaro, Kagera, Dodoma and combination of regions such as (Kagera, Mara and Mwanza), (Mwanza and Shinyanga), (Mwanza and Kagera) and (Kagera, Mara, Shinyanga and Mwanza). Sample size was categorized into 3 categories; <100 (small), 100–499 (moderate) and >500 (large), age of the participant into 3 categories; <18, (18 and above) and all ages, schistostosoma species into 2 categories; *S.mansoni* and *S. haematobium*, and diagnostic method into 12 categories; Kato katz, PCR, POC-CCA, Mini FLOTAC, Sedimentation, Microscopy, Direct wet preparation, Urine filtration, Urine centrifugation, CAA, Direct smear and microhematuria. The factors

demonstrating significant heterogeneity was subjected to multivariate subgroup analysis and the amount contributed by each factor or combination of significant factors ($R^2$) was determined.

Additionally, publication bias was examined visually using a funnel plot and the degree of asymmetry was further confirmed by using Egger's regression test. Whereby, it was assumed that the symmetrical distribution of the study effect sizes across the plot indicates the absence of the study bias. In contrast, the asymmetrical distribution reflects the presence of the study bias. All statistical analyses were done in R software version 4.3.0 using the meta prop function under the meta and metaphor package.

# 3 Results

## 3.1 Literature search results

### 3.1.1 Characteristics of the reviewed publications.

A systematic literature search produced 1504 publications ranging from 2013 to 2023, from which only 55 met the inclusion criteria by reporting on the prevalence datasets from disparate regions of Tanzania and hence included in the present review work (Fig 1). Of the included publications, 3 reported prevalence datasets from women, 4 from preschool-aged children, 22 from school-aged children, 3 from both pre and school-aged children, 2 from both children and adults, 2 from children, 1 from HIV-infected children, 1 from psychiatric patients, 1 from adults with TB, 2 from adults with HIV, 1 from street children and orphans, 10 from adults, 3 from the general population, 1

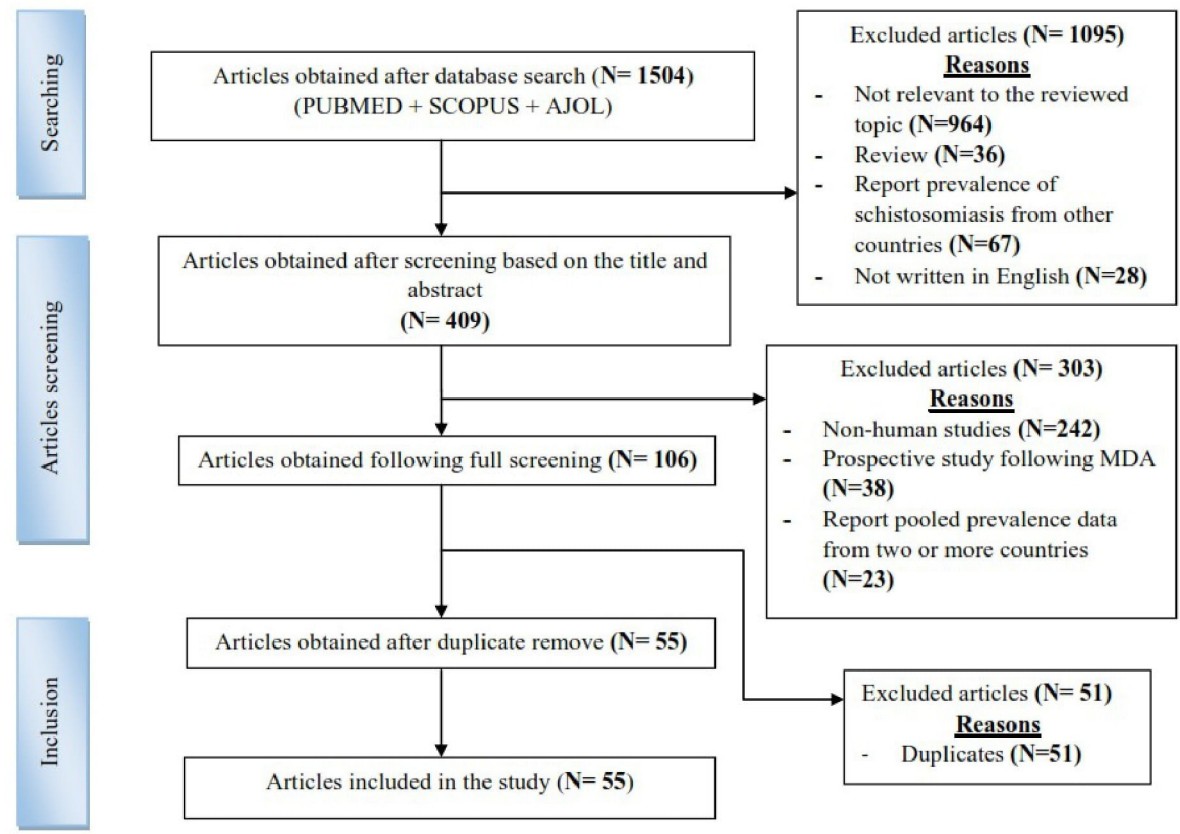

**Fig 1. Conceptual framework of the literature search and screening process.**

from secondary school students and 1 from the university students (Table 1). On the other hand, the study design from the selected 55 publications was 44 cross-sectional studies, 1 case-control, 2 cohorts, 5 prospective longitudinal, 1 randomized controlled trial, and 2 study designs were not specified (Table 1).

Among the selected publications, 30 assessed the prevalence of *S. mansoni*, 15 of *S. hematobium*, and 11 of both *S. hematobium* and *S. mansoni*. The regions from Tanzania were reported as follows; 1 publications report prevalence dataset from Ruvuma, 3 from Lindi, 3 from Mara, 23 from Mwanza, 5 from Dar es Salaam, 2 from Simiyu, 5 from Zanzibar, 2 from Shinyanga, 1 from Geita, 1 from Kagera, 4 from Morogoro, 2 from Dodoma and 1 from Kilimanjaro. Meanwhile, 2 publications report combined datasets from Kagera, Mara, Shinyanga and Mwanza, 1 from Mwanza and Shinyanga and 1 from Mwanza and Kagera (Table 1).

An array of diagnostic methods were used to screen both *S. mansoni* and *S. hematobium* whereby, Kato katz was the mostly used diagnostics method as reported by 30 studies, followed by urine filtration (n = 19), POC-CCA (n = 19), PCR (n = 5), microscopy (n = 4), urine concentration (n = 3), direct smear (n = 2), formal ether concentration (n = 3), CAA (n = 3), microhematuria (n = 1) and Mini FLOTAC (n = 2) (Table 1). The enrolled studies recruited a total of 122,674 individuals from Zanzibar as well as 12 regions of Tanzania mainland. The age of examined individuals ranges from 0 to 95 years.

### 3.2 Quality and risk of bias assessment

Examination of the study quality and risk of bias revealed the absence of low-quality studies; 23 publications had moderate quality hence scored (5–7), whereas 32 studies had high quality with score range of (8–9). The average score was 7.57, which indicates the overall moderate quality of the included publications. All studies used standard procedures of sample collection (stool, urine and blood) as well as valid diagnostic tests for the detection of both urogenital and intestinal schistosomiasis.

### 3.3 Pooled prevalence and sub-group analysis

Fifty-five (55) studies have reported on schistosomiasis prevalence datasets from various regions of Tanzania mainland and Zanzibar, which were used to estimate the country pooled prevalence. The pooled prevalence of schistosomiasis (both urogenital and intestinal) in Tanzania was 26.40% [95% CI: 20.73–32.98, $I^2$ = 98.5%] (Fig 2). Cochran's Q test portrayed substantial heterogeneity at p-value = 0.001, meanwhile, the between study variances determined using Higgins ($I^2$) was also very high (99.6%).

Due to high heterogeneity, sub-group analyses were conducted to assess the effects of the contributing factors, which include Schistosoma species, year of publication, diagnostic methods, sample size, participant's age as well as regions. Sub-group analysis based on the years of publication did not reveal significant heterogeneity at p-value = 0.2716, whereas other assessed factors such as diagnostic methods (p-value = < 0.0001), sample size (p-value = 0.0094), participant's ages (p-value = < 0.0106), regions (p-value = 0.001) and schistosomes specie (p-value = < 0.0001) have shown significant contribution to the observed high heterogeneity. Furthermore, based on the years of publication, the sub-group analysis revealed the pooled prevalence of 23.41% [95% CI: 16.49–32.11] and 30.06% [95% CI: 21.97–39.61] at the following time intervals (2013–2018) and (2018–2023) respectively (Fig 3). Sub-group analysis based on the regions revealed the highest prevalence of schistosomiasis in Mara, Simiyu, and Mwanza with a prevalence of 77.39% [95% CI: 55.86–90.25, $I^2$ = 98.5%], 72.26% [95% CI: 16.70–97.13, $I^2$ = 99.6%], and 51.19% [95% CI: 44.79–57.56, $I^2$ = 98.5%] respectively (Figs 4 and 5). Sub-group analysis of the prevalence dataset based on the Schistosoma species divulged the pooled

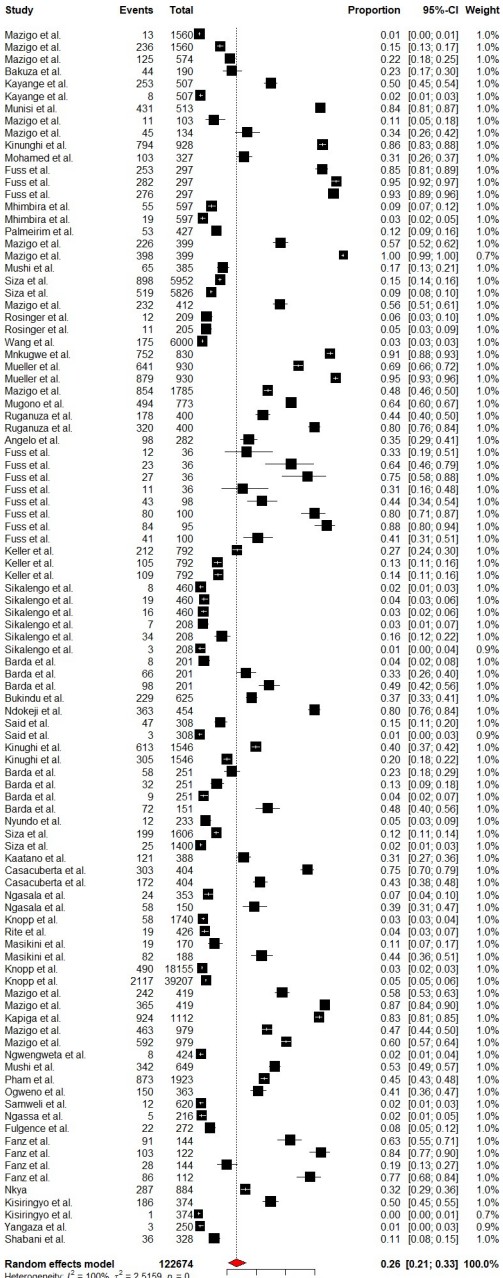

**Fig 2. Forest plot showing the pooled prevalence of schistosomiasis in Tanzania.**

prevalence of 8.86% [95% CI: 5.64–13.65, $I^2$ = 99.5%] and 37.91% [95% CI: 31.05–45.29, $I^2$ = 99.2%] for *S. hematobium* and *S. mansoni* respectively (Fig 6). Furthermore, sub-group analysis based on the diagnostic method showed the highest prevalence of PCR and POC-CCA with a pooled prevalence of 64.11 [95% CI: 31.89–87.21, $I^2$ = 98.4%], 56.45 [95% CI 37.68–73.53, $I^2$ = 99.1%] respectively (Figs 7 and 8). Estimated pooled prevalence based on the sample size sub group were 23.83 [95% CI: 15.59–34.64, $I^2$ = 99.8%], 25.54 [95% CI: 19.38–32.87, $I^2$ = 98.5%] and 0.5795 [95% CI: 0.3670–0.7660, $I^2$ = 91.4%] for the sample size >500, 100–499 and <100 respectively (Fig 9). For the sub group analysis on the basis of the participant ages, the

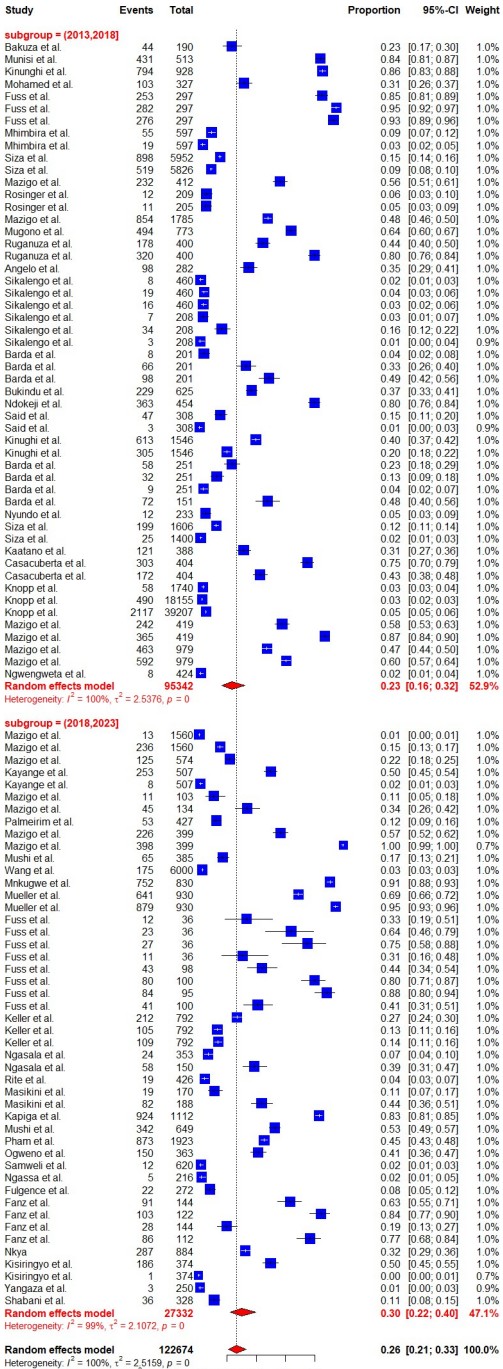

**Fig 3. Forest plot showing the pooled prevalence of schistosomiasis in years of publication subgroups.**

estimated prevalence of schistosomiasis were 29.47 [95% CI: 20.68–40.10, $I^2$ = 99.7%], 15.19 [95% CI: 7.58–28.12, $I^2$ = 99.6%] and 37.21 [95% CI: 27.24–48.40, $I^2$ = 99.0%] for the following age groups; < 18, 18 and above and all ages respectively (Fig 10). The results of multivariate sub group analysis revealed that, factors including age ($R^2$ = 100%), sample size ($R^2$ = 100%), diagnostic method ($R^2$ = 100%) and Region ($R^2$ = 100%) contribute strongly to the overall heterogeneity (variability observed in the effect sizes) (p-value = 0.0001).

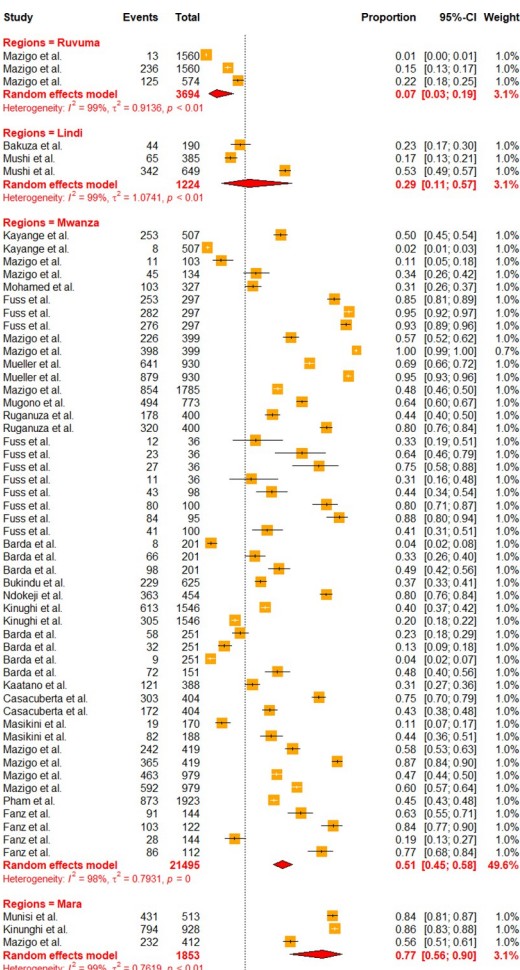

**Fig 4. Forest plot showing the pooled prevalence of schistosomiasis in region sub-group.**

Since Zanzibar and Tanzania mainland have disparate histories regarding schistosomiasis control, additional sub group analysis were conducted to determine the estimated pooled prevalence of the mentioned areas. The estimated prevalence of schistosomiasis in Tanzania mainland was 28.89% [95% CI: 23.61–0.3482, $I^2$ = 99.3%]; meanwhile, for Zanzibar was 8.95% [95% CI 5.11–15.22, $I^2$ = 99.4%] (Fig 11).

## 3.4 Publication bias

Publication bias was assessed using a funnel plot of the effect sizes against standard error; as such, there was a symmetrical distribution of studies effect sizes along the plot, indicating the absence of the publication bias (Fig 12). However, egger's regression analysis for funnel plot asymmetry outcome fails to confirm the absence of publication bias (p-value = 0.0099).

## 4. Discussion

Schistosomiasis (both urogenital and intestinal) is an endemic disease in the Sub-Saharan region and the most devastating parasitic disease behind Malaria. Tanzania, being located in this region, is also highly affected by schistosomiasis. The present review assessed the prevalence of schistosomiasis in various regions of Tanzania mainland and Zanzibar and reported

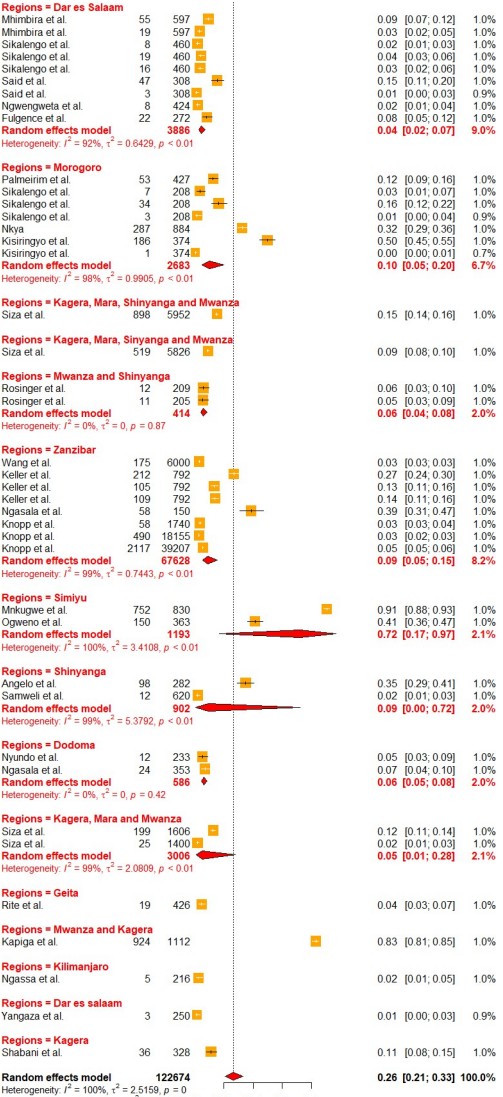

**Fig 5. Forest plot showing the pooled prevalence of schistosomiasis in region sub-group.**

their pooled estimate values (prevalence). To the best of our knowledge, this is the first review reporting the prevalence of schistosomiasis in Tanzania for the past ten years (2013–2023).

The meta-analysis of the dataset from included studies revealed a high pooled prevalence of Schistosomiasis in Tanzania (26.40%). Notably, the prevalence was higher in Tanzania mainland than in Zanzibar. The lower prevalence in Zanzibar can be attributed to the successful implementation of the praziquantel mass drug administration strategy, which was carried out biannually for over 6 years [12,13]. Other strategies, such as behavioural changes and bio-control of the disease intermediate host, also contributed to the reduction of the disease burden [12,13]. However, the high prevalence in the mainland suggests that some regions may not have responded effectively to the control interventions or were not sufficiently involved in the treatment strategies, leading to the overall rise in prevalence.

The degree of heterogeneity and between study variances were high and attributed to factors including sample size, regions, diagnostic method, schistosoma species, and participant

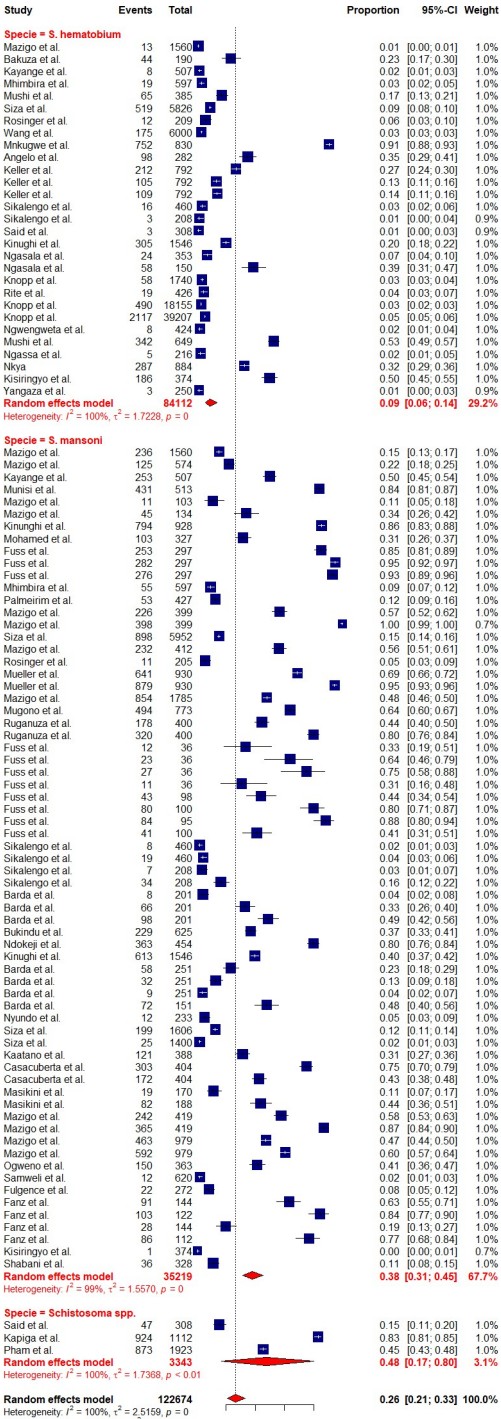

**Fig 6. Forest plot showing the pooled prevalence of schistosomiasis based on specie sub-group.**

age. The prevalence of intestinal Schistosomiasis was high compared to urogenital schistosomiasis. The high prevalence of intestinal schistosomiasis can be attributed to the tendency of people to defecate near water bodies due to the presence of long vegetation and the availability of water for cleaning themselves, as was previously reported by Zacharia and coworkers (2020). This, therefore, fosters the spread of the disease through exposure of susceptible

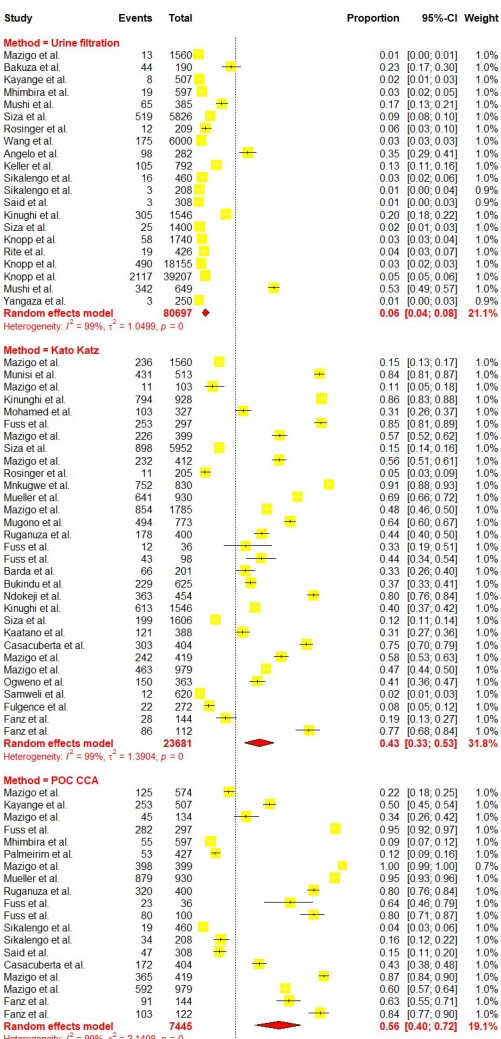

**Fig 7. Forest plot showing the pooled prevalence of schistosomiasis based on the diagnostic method.**

individuals to the infected water. The observed high prevalence of intestinal schistosomiasis is supported by the study done by Zacharia and coworkers (2020), which reported the re-infection rate and the prevalence of intestinal schistosomiasis at the global scale being higher relative to that of urogenital schistosomiasis [17]. The five-year interval analysis of schistosomiasis prevalence revealed a significant increase. As explained above, this surge may indicate inconsistency and lack of sustainability in the implementation of disease control interventions, as some regions might not have effectively responded to or have ceased implementing the treatment strategies, leading to a rise in overall prevalence. The north-western zone, which comprises regions including Mara, Simiyu, and Mwanza, has a pronounced pooled prevalence of the disease. The observed high prevalence could be affiliated with the presence of Lake Victoria, which is a potential source of transmission, especially when people conduct their socioeconomic activities such as fishing, agriculture, laundry, bathing, and mining, among others, thereby increasing the risk of subsequent re-infection [10].

On the other hand, proper and timely detection of schistosomiasis is an imperative step toward the efficient elimination of the disease. Failure to correctly diagnose the disease will, therefore, compromise the intensive work conducted to control the disease. Sub-group

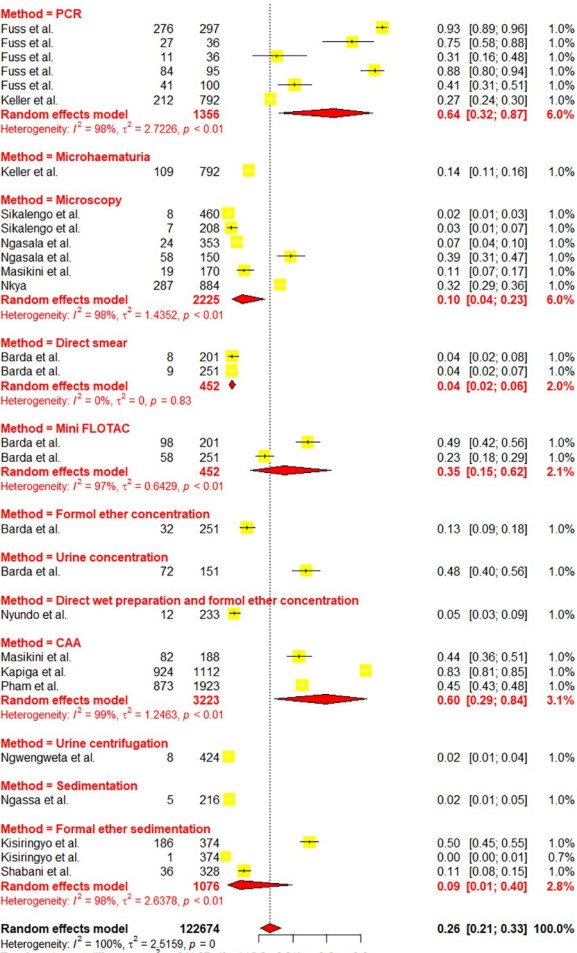

**Fig 8. Forest plot showing the pooled prevalence of schistosomiasis based on the diagnostic method.**

analysis based on the diagnostic methods divulged the high prevalence of schistosomiasis detected by PCR and POC-CCA. This indicates that the aforementioned methods are highly sensitive and, indeed, effective for the detection of both urogenital and intestinal schistosomiasis. As such, they should be widely adopted in the detection of schistosomiasis, as recommended by Bisetegn and coworkers, 2021 [72]. For sub group analysis on the basis of the participant age, it was revealed that participants of all ages were the most infected group. The high prevalence of this age group is because it encompasses all vulnerable groups, including school aged children, women of reproductive age, fishermen, etc., whose probability of exposure to infected water is relatively high. This observation agrees with another study that reported a high re-infection rate of schistosomiasis in individuals of all ages compared to other age groups [17]. Studies with small sample sizes had a high prevalence compared to those with moderate and large sample sizes. The reason behind the high prevalence of schistosomiasis in studies with small sample sizes could be the easy and intensive follow up compared to the ones with large sample sizes. Visual examination of the funnel plot portrayed the absence of the study bias as the study effect sizes were symmetrically distributed across the plot. However, eggers regression test outcomes failed to confirm the absence of the plot asymmetry; hence, it indicates the potential missing out of some studies, particularly the ones with small effect sizes.

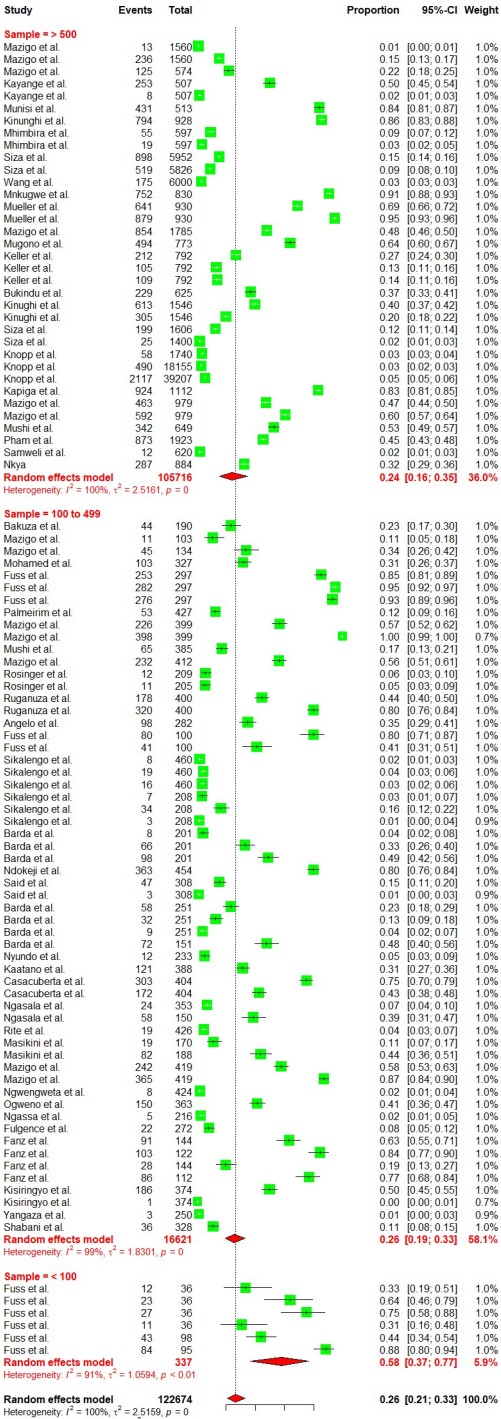

**Fig 9. Forest plot of the sub group analysis based on the sample size.**

## 5. Conclusion and recommendation

Despite extensive efforts put in place by the government of Tanzania in trying to eliminate schistosomiasis as a complement to the sustainable development goal number 3 of attaining health and wellbeing for all by 2030, the disease prevalence is still growing, as stipulated in this

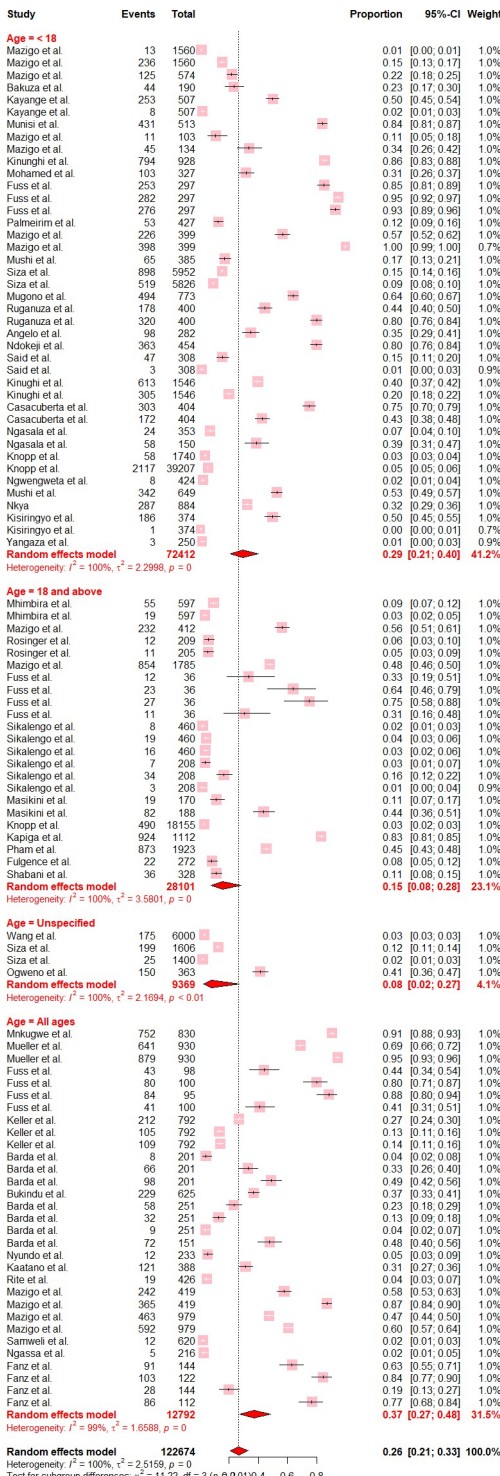

**Fig 10. Plot of the sub group analysis based on the participants ages.**

review. The regions surrounding Lake Victoria had the higher pooled prevalence and were considered the core hub for subsequent re-infection of schistosomiasis. The review also highlights that *S. mansoni* is the most prevalent species in Tanzania relative to *S. hematobium*. In

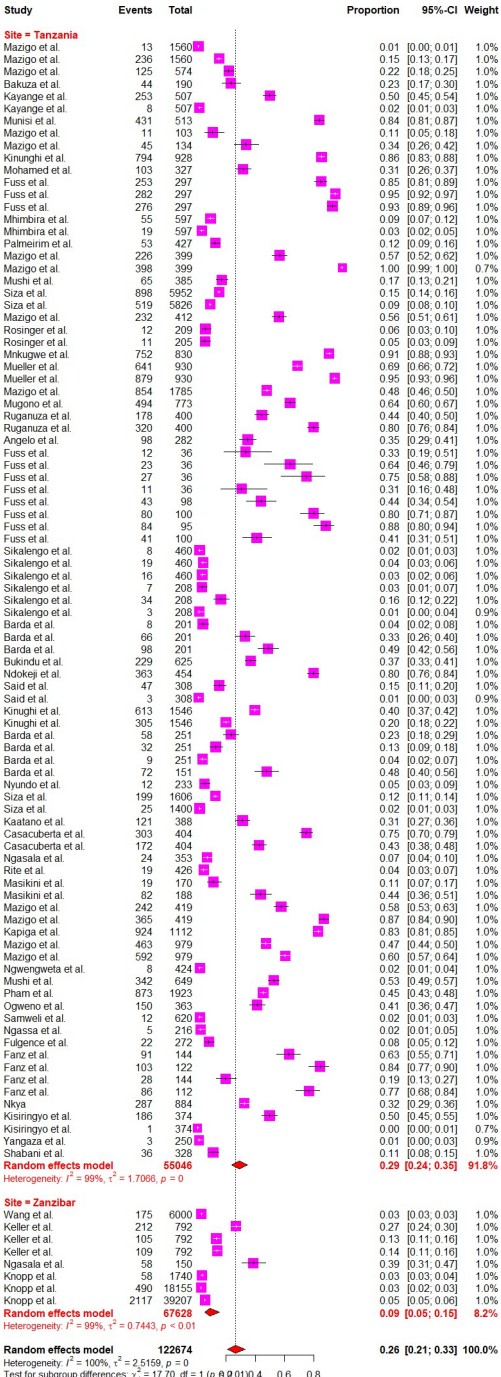

**Fig 11. Forest plot showing the pooled prevalence of schistosomiasis in Tanzania mainland and Zanzibar.**

view of that, an intensive deployment of praziquantel mass drug administration in combination with other control strategies in the endemic regions, particularly in the mainland part, is of paramount importance. The present study also proves that PCR and POC-CCA are the most sensitive methods for the detection of both urogenital as well as intestinal schistosomiasis as compared to the commonly used microscopy and direct smear methods. Given this, the

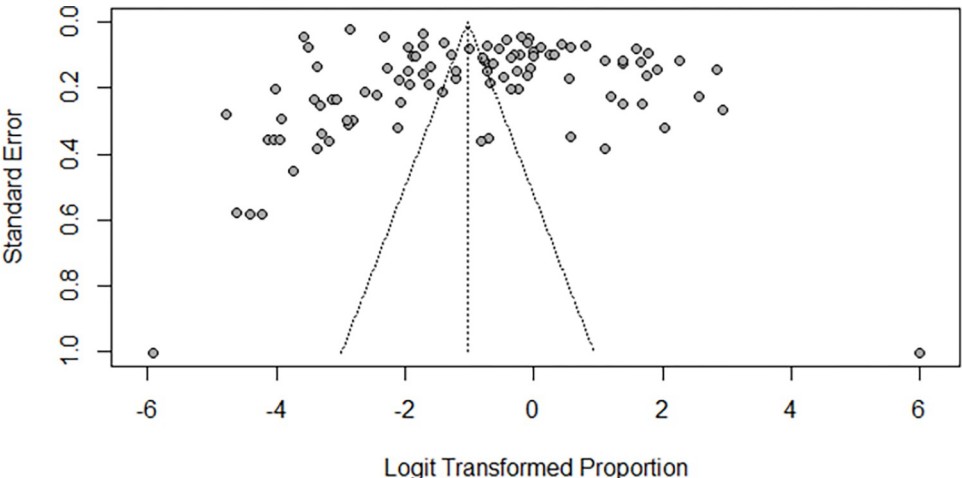

**Fig 12. Funnel plot showing the symmetrical distribution of studies along the plot.**

aforementioned methods should be adopted during disease management interventions as the methods of choice for the identification of disease cases.

## Acknowledgments

The authors acknowledge the management of the Nelson Mandela African Institution of Science and Technology for their administrative support, particularly by facilitating us with the office space, internet services, and other infrastructure needed during the whole process of writing this systematic review and meta-analysis.

## Author Contributions

**Conceptualization:** Nicolaus Omari Mbugi.

**Data curation:** Nicolaus Omari Mbugi, Hudson Laizer.

**Formal analysis:** Nicolaus Omari Mbugi, Hudson Laizer.

**Methodology:** Nicolaus Omari Mbugi.

**Supervision:** Musa Chacha, Ernest Mbega.

**Writing – original draft:** Nicolaus Omari Mbugi.

**Writing – review & editing:** Hudson Laizer, Musa Chacha, Ernest Mbega.

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
