## [Decision Letter · Decision Letter 0]

8 Feb 2024

Dear Mr Mbugi,

Thank you very much for submitting your manuscript "Prevalence of Human Schistosomiasis in Various Regions of Tanzania Mainland and Zanzibar: A Systematic Review and Meta-Analysis of Studies Conducted for the Past Ten Years (2013-2023)" for consideration at PLOS Neglected Tropical Diseases. As with all papers reviewed by the journal, your manuscript was reviewed by members of the editorial board and by several independent reviewers. In light of the reviews (below this email), we would like to invite the resubmission of a significantly-revised version that takes into account the reviewers' comments. 

We cannot make any decision about publication until we have seen the revised manuscript and your response to the reviewers' comments. Your revised manuscript is also likely to be sent to reviewers for further evaluation.

Sincerely,

Uwem Friday Ekpo, PhD

Academic Editor

Francesca Tamarozzi

Section Editor

Reviewer's Responses to Questions

**Key Review Criteria Required for Acceptance?**

**Methods**

-Are the objectives of the study clearly articulated with a clear testable hypothesis stated?

-Is the study design appropriate to address the stated objectives?

-Is the population clearly described and appropriate for the hypothesis being tested?

-Is the sample size sufficient to ensure adequate power to address the hypothesis being tested?

-Were correct statistical analysis used to support conclusions?

-Are there concerns about ethical or regulatory requirements being met?

Reviewer #1: (No Response)

Reviewer #2: (No Response)

**Results**

-Does the analysis presented match the analysis plan?

-Are the results clearly and completely presented?

-Are the figures (Tables, Images) of sufficient quality for clarity?

Reviewer #1: (No Response)

Reviewer #2: (No Response)

**Conclusions**

-Are the conclusions supported by the data presented?

-Are the limitations of analysis clearly described?

-Do the authors discuss how these data can be helpful to advance our understanding of the topic under study?

-Is public health relevance addressed?

Reviewer #1: (No Response)

Reviewer #2: (No Response)

**Editorial and Data Presentation Modifications?**

Reviewer #1: (No Response)

Reviewer #2: (No Response)

**Summary and General Comments**

Reviewer #1: Here, the authors present a review and meta-analysis of literature on prevalence of schistosomiasis across Tanzania. This is a valuable contribution to the literature however it requires a major revision to be worthy of publication. Overall there is a noticeable lack of proper citation. Either there are no citations used where there should be, or non-primary sources are used. There are also lots of grammatical errors. Finally, I think the authors are rather over egging the pudding when it comes to the importance of this work. As I have previously said, this is an important contribution, however there is an overuse of journalistic exaggeration that weakens the narrative. 

My specific comments are below: 

Make your code openly available. 

- Species not specie. 

- Please use line numbers to make reviewing easier. 

- The final paragraph of the abstract, S. mansoni is not defined anywhere in the abstract prior to this. My initial thought is that the abstract is actually too long. Much of the detail could be exchanged for a more general overview because as it stands there are lots of details that without greater context do not necessarily make sense. 

Introduction

- Schistosomiasis is not a pandemic. It is a tropical and sub-tropical disease that disproportionately infects those in resource limited settings. Even if we consider the small number of cases in Corsica, these were isolated, and transmission has died out. 

- You say five species of clinical importance and then list six. Here, you can also use Schistosoma species name for the first one in the list and then S. species name for the following five. 

- 5 (which should be 6) should be written out (five) as it is less than 10. 

- What do you mean the “former 4” – they have not changed, I think you mean the first five cause intestinal disease? 

- The mortality rates are not second after malaria. The socio-economic impact is second only to malaria. You have also not cited a primary source for this statement. 

- Please use primary sources, again the article cited when stating that TZ is only surpassed by Nigeria for prevalence is not the primary source. The primary source is a 2009 piece by Hoetz et al. I should not have to go digging for primary sources and I would also suggest given the work of the SCI there are likely more up to date estimates available now. 

- The statement “the country is endemic to…” is incorrect, it should be the other way around “S. mansoni and S. haematobium are endemic in Tanzania”. 

- Adult worms do not contribute to morbidity. If someone has an entire single sex population, if there are no eggs there is no morbidity – as per the controlled human infections happening in the Netherlands. 

I think section 2.3 should be data EXTRACTING not abstracting. 

- “Meta-analysis was done in line with the previously published protocol by employing a random effects model.“ where is the citation for this if it is previously published? I believe Ayabina, Clark et al PLOS NTDS uses the same methodology but I am not sure whether this is the source here. 

- Are meta and metaphor two separate packages? Citations please. 

3.1 Characteristics of the reviewed publications 

- I would like the citations for each of these descriptors. For example, the 3 papers that reported prevalence datasets from women, what are the citations for these? I wonder if a table would be a better way of presenting this information – though I think table 2 is too much. I would put these into your categories that you’re presenting in the first paragraph of this section. 

- Same for the second paragraph, I wonder if you want to produce a map or something to summarise these locations. There is a tendency to focus epi research in the north of Tanzania, or on Zanzibar – I think a visual representation of this would be quite powerful. 

Section 3.2 

- What do you mean “several studies were pooled” – this is very woolly. Which studies? Why did you pool them? Be more explicit here. 

- Take out “whereby,” it doesn’t make sense. 

- What do you mean concurrently? This doesn’t make sense either. 

- Your pooled time periods both include 2018, do your estimated prevalence both include 2018 also? This should not be this way. 2013-2017 and 2018-2023 or 2013-2018 and 2019-2023 should be used. 

- What do you mean on the other hand? As opposed to what? 

- In figure 1 why is there a dotted line just above 0.2? Is this the overall mean? 

- What is the publication bias in comparison to? Spatially I am sure there is a bias. 

Discussion

- “as regards to general disease sequelae” doesn’t make sense. – I would actually get rid of the first four lines of the discussion. There are no citations, it is repetitive of the introduction, and it is poorly written. Focus on the findings, rather than trying to make an unsubstantiated point to make the work seem more important. 

- The second paragraph of the discussion lacks direction – what is the point that the authors are trying to make?

- Stop using journalist language like “alarming” – just report on what you have and put it into the context of the literature without fear mongering. 

- “The high prevalence observed indicates that the rate of reinfection is still very high despite the implementation of several rounds of disease control interventions” There is no description of the intervention programmes underway in TZ, nor any citations to support this statement. If the authors want to make the point that treatment programmes aren’t working (which I have no doubt they aren’t based on results from other SSA countries) then there needs to be a more description of the activities that are in place. 

- “As a consequence, there is alarming potential existence of a resistant strain of S. mansoni circulation amongst our population.” The work you mention (but do not properly cite) is not evidence for resistance. There is very little evidence to support PZQ resistance (see work by D. Berger, C. Faust and others). 

- “This warrants the use of new and most effective strategies preferably the multifaceted approach” – what are the most effective strategies? What is THE multifaceted approach? There are no citations here and no concrete examples of what these interventions should be. I feel like the “conclusions” part would actually make a good opening to your discussion. 

- Your review provides ZERO evidence of treatment resistance, this is a dangerous message to spread when there is so little evidence to support this hypothesis in the first place. In reality, there are so few people treated in the grand scheme of transmission, that the refuge population of worms is probably larger than those exposed to treatment, such that the selective pressure to generate resistance just is not there. 

- I don’t understand why there is a section 5. Surely the normal format is that the discussion contains conclusions and recommendations. 

- I fundamentally disagree that POC-CCA is the best for urogenital infection (for example https://doi.org/10.1179/136485908X337490)

Reviewer #2: I have presented my comments in the attached document.

PLOS authors have the option to publish the peer review history of their article (what does this mean?). If published, this will include your full peer review and any attached files.

Reviewer #1: No

Reviewer #2: No
---

## [Decision Letter · Decision Letter 1]

20 Apr 2024

Dear Mr Mbugi,

Thank you very much for submitting your manuscript "Prevalence of Human Schistosomiasis in Various Regions of Tanzania Mainland and Zanzibar: A Systematic Review and Meta-Analysis of Studies Conducted for the Past Ten Years (2013-2023)" for consideration at PLOS Neglected Tropical Diseases. As with all papers reviewed by the journal, your manuscript was reviewed by members of the editorial board and by several independent reviewers. The reviewers appreciated the attention to an important topic. Based on the reviews, we are likely to accept this manuscript for publication, providing that you thoroughly modify the manuscript according to all the review recommendations. Please do taje oarticular care in shortening the text, as required, and avoid overinterpretation of results or misleading conclusions.

Sincerely,

Uwem Friday Ekpo, PhD

Academic Editor

Francesca Tamarozzi

Section Editor

Reviewer's Responses to Questions

**Key Review Criteria Required for Acceptance?**

**Methods**

-Are the objectives of the study clearly articulated with a clear testable hypothesis stated?

-Is the study design appropriate to address the stated objectives?

-Is the population clearly described and appropriate for the hypothesis being tested?

-Is the sample size sufficient to ensure adequate power to address the hypothesis being tested?

-Were correct statistical analysis used to support conclusions?

-Are there concerns about ethical or regulatory requirements being met?

Reviewer #1: (No Response)

Reviewer #2: (No Response)

**Results**

-Does the analysis presented match the analysis plan?

-Are the results clearly and completely presented?

-Are the figures (Tables, Images) of sufficient quality for clarity?

Reviewer #1: (No Response)

Reviewer #2: (No Response)

**Conclusions**

-Are the conclusions supported by the data presented?

-Are the limitations of analysis clearly described?

-Do the authors discuss how these data can be helpful to advance our understanding of the topic under study?

-Is public health relevance addressed?

Reviewer #1: (No Response)

Reviewer #2: (No Response)

**Editorial and Data Presentation Modifications?**

Reviewer #1: (No Response)

Reviewer #2: (No Response)

**Summary and General Comments**

Reviewer #1: - The abstract is way too long and contains unnecessary information. At most I would expect an abstract to be ~300 words. You have four paragraphs. That is not an overview that is just your methods copied and pasted. For example:

- Remove “The prevalence datasets from the included studies were analyzed by using R software version 4.3.0” 

- Remove “via meta prop function under the meta and metafor package”

Lines 42-43: What do you mean a time-dependent increase? You just mean it is going up? Do not try to sound fancy, just tell me what your data show. And how does that show an increasing rate of reinfection specifically? Surely it shows an increase in the force of infection? 

Line 45: Species. 

Line 47: I said this last time you submitted this article and I say it again. There is practically no evidence to support the statement that there is emerging treatment resistance in S. mansoni. Remove this statement. It is not helpful to management programmes and is fundamentally untrue. Consider the ecology and evolution of S. mansoni – there is a massive refugia because so many people are untreated and because there are so many thousands of cercaria produced. The selective pressure to force resistance into fixation in a population is just not there. Genomic evidence supports this. Stop using this trope. This goes for its mention in the discussion too. 

Is it not the case that Tanzania has stopped the schistosomiasis programme now? Perhaps this deserves a mention in the introduction as I fear it is a massive mistake. 

Line 239: Remove “generally” – you have reported the exact number of participants across all studies. There is nothing general about that. 

Table 1 - Personally, I would put this table in the supplementary material. 

I am interested to know more about your “quality” assessment. Personally, I think there is a tendency for poor statistics in schistosomiasis research and I would like to know if this was a metric in your assessment. 

Line 255 remove “using the metaprop function” this does not really mean anything to someone unfamiliar with the package and should be in your methods not your results. 

Lines 264 – theres no such thing as a p value of 0. 

Line 342 – I am still confused by your use of “high rates of reinfection” – it is the force of infection that is high surely. And if you are claiming that treatment is not working because of resistance, then people are not getting reinfected, they are still infected. Whilst I can imagine that reinfection rates are high in areas like Mwanza where contact with contaminated water is constant, I do not agree that overall your results show high rates of reinfection. 

340 to 366 – this paragraph is enormous and meandering. What is your point? 

Lines 354 – 356 – I have no idea what point you are trying to make here. 

Line 375 you say “participants of all ages” then go on in the following to say “this age group”. What age group, you have just said all age groups. 

I feel like your discussion just repeats a lot of your results. This makes your discussion incredibly repetitive and long and quite boring to read. I would like to see how your results are fitting in with the wider literature and what this all means for control.

Reviewer #2: (No Response)

PLOS authors have the option to publish the peer review history of their article (what does this mean?). If published, this will include your full peer review and any attached files.

Reviewer #1: No

Reviewer #2: No

Figure Files:

Data Requirements:

Reproducibility:

References

---

## [Decision Letter · Decision Letter 2]

5 Jul 2024

Dear Mr Mbugi,

Thank you very much for submitting your manuscript "Prevalence of Human Schistosomiasis in Various Regions of Tanzania Mainland and Zanzibar: A Systematic Review and Meta-Analysis of Studies Conducted for the Past Ten Years (2013-2023)" for consideration at PLOS Neglected Tropical Diseases. As with all papers reviewed by the journal, your manuscript was reviewed by members of the editorial board and by several independent reviewers. The reviewers appreciated the attention to an important topic. Based on the reviews, we are likely to accept this manuscript for publication, providing that you modify the manuscript according to the review recommendations. 

Sincerely,

Uwem Friday Ekpo, PhD

Academic Editor

jong-Yil Chai

Section Editor

Reviewer's Responses to Questions

**Key Review Criteria Required for Acceptance?**

**Methods**

-Are the objectives of the study clearly articulated with a clear testable hypothesis stated?

-Is the study design appropriate to address the stated objectives?

-Is the population clearly described and appropriate for the hypothesis being tested?

-Is the sample size sufficient to ensure adequate power to address the hypothesis being tested?

-Were correct statistical analysis used to support conclusions?

-Are there concerns about ethical or regulatory requirements being met?

Reviewer #2: (No Response)

Reviewer #3: The methods were carried out in line with a systematic review and were very informative and easy to follow. The fact that most authors were included in the review process is excellent in avoiding subjective choices.

I have some minor comments for the methods: 

In the exclusion and inclusion criteria I would like to have understood the justification for two points:

• Not including studies which included data sets from longitudinal studies following mass drug administration - what is the reason for this? Could you not have included any baseline data from these studies?

• Studies assessing genetic dynamics among S. haematobium populations – again can you explain why this is. Furthermore, I would like to remind the authors that to avoid reviewer bias all inclusion and exclusion criteria should be set before any reviewing takes place, therefore surely this exclusion criteria should include all species of Schistosoma unless there is a specific reason why S. haematobium should be excluded and others not.

Can the authors explain why they chose to subgroup the publication years as they did, is there specific justification for splitting them into a five-year period? Was the number of studies too low to be include each year separately?

Finally, can the authors explain how they ensured that articles were not using the same datasets, it is not uncommon for a lab group to use the same dataset with different focus but both reporting prevalence. Please can the authors state how they ensured they were not duplicating data here. And add this to the exclusion criteria.

**Results**

-Does the analysis presented match the analysis plan?

-Are the results clearly and completely presented?

-Are the figures (Tables, Images) of sufficient quality for clarity?

Reviewer #2: (No Response)

Reviewer #3: Table 1 – Can this table be ordered by either author name (alphabetically) or year or sample size? This would make it easier to read.

I was surprised to not see any articles with first author of either Pennance or Trippler in this table as they have both carried out extensive work in Zanzibar, but perhaps this is due to the reason above – that they are using datasets from previous studies by Knopp et al. If this was made clear in the exclusion criteria it would not be so surprising that the articles by these authors are not in the table. Otherwise, can the authors check that articles by these authors should not have been included.

In the subgroup analysis it would have been informative to see prevalence stratified by region and parasite together so we can really see what is happening in each region. As some regions have very low eg S. mansoni and very high S. haematobium the pooled prevalence would not be very informative as will be in the middle of the two.

The forest plots are well laid out and easy to read.

**Conclusions**

-Are the conclusions supported by the data presented?

-Are the limitations of analysis clearly described?

-Do the authors discuss how these data can be helpful to advance our understanding of the topic under study?

-Is public health relevance addressed?

Reviewer #2: (No Response)

Reviewer #3: Line 346 states that the high prevalence of S. mansoni prevalence can be attributed to the tendency of people to defecate near water bodies – where in this review did the authors find this? If this was from other studies, please be clear of this.

Line 352 – the authors should be careful with what they say their results show – an increase of pooled prevalence from all parasite species over all regions does not necessarily show an inefficiency of PZQ by MDA as a disease control and elimination strategy. As this is pooled prevalence, it could be that some regions have responded very well to MDA, but this is diluted by other regions which either had not been involved in treatment strategies or had low coverage due to unknown reasons – but we cannot know this from the data presented here. Furthermore, some regions will have had additional control strategies such as snail control etc. but this is not accounted for in the results. The authors themselves state that the control interventions initiated by ZEST in the islands has reduced disease burden from above 50% to below 5%, therefore contradicting the aforementioned claim that control interventions are not working in the country.

**Editorial and Data Presentation Modifications?**

Reviewer #2: (No Response)

Reviewer #3: (No Response)

**Summary and General Comments**

Reviewer #2: The authors have addressed the comments raised in my previous review report, and I see significant improvement. However, before being accepted, I recommend that the manuscript be revised by a professional English language editor.

Reviewer #3: The introduction was particularly interesting and covered the history of control strategies in the country well. I felt very informed and ready to read the rest of the article by the end of the introduction.

One comment for the intro:

Line 73 – High prevalence does not necessarily mean high morbidity, the authors say Tanzania marks the second country in terms of high disease morbidity rate, however neither reference cited here support this claim. Reference 6 states that there is significant morbidity in Tanzania, but they cite a different study to support this and do not report any morbidity results themselves. Reference 68 does not specifically talk about morbidity in Tanzania.

Overall summary:

My only real issue with this article is that I do not agree with the authors claim of what it shows the results can be used for. Line 128 they state that the results from this article can be used as a baseline roadmap for the proper allocation of resources and in the discussion line 333 they state that the observed prevalence indicates that the force of infection is still very high. I find it difficult to see how pooled prevalence of ten years tells us anything about what is happening at the moment, or can be used as a roadmap to inform future strategies. I can see how the data contained in this article could be very useful in informing control strategies if they had looked at how prevalence had changed in response to control strategies over the years. But a pooled prevalence from many different years and regions and parasite species does not allow us to inform on the current situation. Reporting of this pooled prevalence is interesting but does not inform in the way the authors state it does. 

In conclusion, this is a well written and well carried out systematic review of the prevalence of Schistosoma over a ten-year period. However, I think some of the claims made by the authors of what these data can actually show are not founded in the way the results are presented.

PLOS authors have the option to publish the peer review history of their article (what does this mean?). If published, this will include your full peer review and any attached files.

Reviewer #2: No

Reviewer #3: No

Figure Files:

Data Requirements:

Reproducibility:

References

---

## [Decision Letter · Decision Letter 3]

17 Aug 2024

Dear Mr Mbugi,

We are pleased to inform you that your manuscript 'Prevalence of Human Schistosomiasis in Various Regions of Tanzania Mainland and Zanzibar: A Systematic Review and Meta-Analysis of Studies Conducted for the Past Ten Years (2013-2023)' has been provisionally accepted for publication in PLOS Neglected Tropical Diseases.

Best regards,

Uwem Friday Ekpo, PhD

Academic Editor

Jong-Yil Chai

Section Editor

Reviewer's Responses to Questions

**Key Review Criteria Required for Acceptance?**

**Methods**

-Are the objectives of the study clearly articulated with a clear testable hypothesis stated?

-Is the study design appropriate to address the stated objectives?

-Is the population clearly described and appropriate for the hypothesis being tested?

-Is the sample size sufficient to ensure adequate power to address the hypothesis being tested?

-Were correct statistical analysis used to support conclusions?

-Are there concerns about ethical or regulatory requirements being met?

Reviewer #3: Yes

**Results**

-Does the analysis presented match the analysis plan?

-Are the results clearly and completely presented?

-Are the figures (Tables, Images) of sufficient quality for clarity?

Reviewer #3: Yes

**Conclusions**

-Are the conclusions supported by the data presented?

-Are the limitations of analysis clearly described?

-Do the authors discuss how these data can be helpful to advance our understanding of the topic under study?

-Is public health relevance addressed?

Reviewer #3: Yes

**Editorial and Data Presentation Modifications?**

Reviewer #3: None

**Summary and General Comments**

Reviewer #3: I think this is a very interesting and well presented research article. Thank you for answering my previous questions and actioning my suggestions. I was a little disappointed that my suggestion to see a sub group analysis of species and region together was not actioned. I do think this would be a more informative way to understand the pooled prevalence in each region (stratified by species), and I would have been interested to see these results. However, as this is one point and is not imperative to the article results, and the overall paper is very good, I decided not to push it and to accept the article as it is. Thank you for spending the time doing this meta-analysis, it is very obvious you have put in a huge amount of work into this.

PLOS authors have the option to publish the peer review history of their article (what does this mean?). If published, this will include your full peer review and any attached files.

Reviewer #3: No

---

## [Editor Report · Acceptance letter]

25 Aug 2024

Dear Mr Mbugi,

We are delighted to inform you that your manuscript, "Prevalence of Human Schistosomiasis in Various Regions of Tanzania Mainland and Zanzibar: A Systematic Review and Meta-Analysis of Studies Conducted for the Past Ten Years (2013-2023)," has been formally accepted for publication in PLOS Neglected Tropical Diseases.

Best regards,

Shaden Kamhawi

co-Editor-in-Chief

Paul Brindley

co-Editor-in-Chief
